# Systematic analysis of crystalline phases in bosonic lattice models with algebraically decaying density-density interactions

Jan A. Koziol[1★] ⓘ, Antonia Duft[1] ⓘ, Giovanna Morigi[2] ⓘ, and Kai P. Schmidt[1†] ⓘ,

**1** Department of Physics, Friedrich-Alexander-Universität Erlangen-Nürnberg,
Staudtstraße 7, Germany
**2** Theoretical Physics, Saarland University, Campus E2.6, D-66123 Saarbrücken, Germany

★ jan.koziol@fau.de , † kai.phillip.schmidt@fau.de

## Abstract

We propose a general approach to analyse diagonal ordering patterns in bosonic lattice models with algebraically decaying density-density interactions on arbitrary lattices. The key idea is a systematic search for the energetically best order on all unit cells of the lattice up to a given extent. Using resummed couplings we evaluate the energy of the ordering patterns in the thermodynamic limit using finite unit cells. We apply the proposed approach to the atomic limit of the extended Bose-Hubbard model on the triangular lattice at fillings $f = 1/2$ and $f = 1$. We investigate the ground-state properties of the antiferromagnetic long-range Ising model on the triangular lattice and determine a six-fold degenerate plain-stripe phase to be the ground state for finite decay exponents. We also probe the classical limit of the Fendley-Sengupta-Sachdev model describing Rydberg atom arrays. We focus on arrangements where the atoms are placed on the sites or links of the Kagome lattice. Our method provides a general framework to treat cristalline structures resulting from long-range interactions.


# 1   Introduction

Long-range interactions[1] play a crucial role in the description of many single and many-body problems [1–5]. The most fundamental examples are the gravitational and the electromagnetic interactions, whose two-body potential decays algebraically with $1/r$ [1–5]. It is therefore natural that effective interactions between neutral objects composed by charged particles, such as molecules and atoms, display long-range interactions [1–5].

Regarding research in the field of statistical mechanics and quantum many-body physics, most phenomena are explained with models using interactions that decay faster than algebraic, e. g., exponentially or short-range interactions,[2] as the fundamental electromagnetic interactions are considered to be screened [6,7]. Long-range interactions, on the other hand, can induce long-range order and their interplay with external potentials gives rise to interesting dynamics and structures. A systematic characterization of the emerging static structures is relevant to several prominent problems in many-body physics.

An example of classical many-body physics [8–11] is melting in two dimensions, where the phase transition changes form the one of hard-spheres [12] to a double Kosterlitz-Thouless-Halperin-Nelson-Young scenario [13–15] for two-body potentials $V(|\vec{r}|) \propto |\vec{r}|^{-\alpha}$ with decay exponents $\alpha \lessgtr 6$ [11]. The phase diagram of clean colloidal systems is highly sensitive to substrate potentials with the possibility of solid superstructures and modulated solid phases in which $n$-mers solidify at the minima of the potential [16–18]. For certain fractional fillings superstructures of minima with different fillings may arise [19, 20]. Long-range interacting colloidal systems on a substrate potential can be seen as a classical continuum extension to the lattice problems analysed within this work.

Another prominent example are magnetic systems. For ferromagnetic lattice models the critical behaviour of the ferromagnetic-paramagnetic phase transition is continuously changing for decay exponents $\alpha < d + 2 - \eta_{\mathrm{SR}}$,[3] from the short-range behaviour towards a long-range mean-field regime [21–25]. For antiferromagnetic lattice models a large emphasis lays on systems where the short-range limit experiences geometrical frustration, as long-range interactions introduce a hierarchy of additional length scales which potentially break the extensive gound-state degeneracy arising from the geometrical frustration and stabilise a certain ground-state pattern. An example for this mechanism is present for the long-range Ising model on triangular lattice cylinders, where the extensive ground-state degeneracy in the nearest neighbour limit is lifted by the long-range interactions and stripe orders emerge as the ground state [26].

---

[1]Interaction potentials that decay asymptotically with $\frac{1}{r^\alpha}$ for the distance $r \to \infty$ with $\alpha \in \mathbb{R}_+$.

[2]Interaction potentials with a finite range such that $\exists \bar{r} \ni V(r > \bar{r}) = 0$. Note that long-range interactions in the limiting case $\alpha \to \infty$ are also short-range interactions.

[3]We define $d$ as the dimension of the system and $\eta_{\mathrm{SR}}$ as the anomalous dimension of the short-range transition.

In ferromagnetic quantum many-body lattice systems analogous observations were recently reported regarding the change in the nature of the critical point [27–34]. For antiferromagnetic interactions it has been demonstrated that long-range interactions generically reduce the stability of a symmetry-broken phase [26,28,30,32]. On frustrated lattices long-range interactions may lead to modifications to the quantum phase diagram with respect to the nearest-neighbour limit as the hierarchy of interactions may break mechanisms such as the order-by-disorder scenario by lifting extensive degeneracies [26,30,35]. Note that spin-1/2 degrees of freedom can be mapped exactly into a hardcore bosonic language using a Matsubara-Matsuda transformation [36].

Important experimental platforms for the realisation of long-range interacting lattice models in a controlled environment are, among others, trapped ions interacting with optical potentials [37–49] and neutral atoms [50–66] as well as polar molecules [67–69] in optical lattices. Both platforms can realise effective long-range Ising or $XY$-type spin interactions, making them viable quantum simulators for long-range magnetic models. Neutral atoms and polar molecules realise fixed decay exponents, while the decay exponent can be tuned in trapped-ion systems. Despite the usage of trapped atomic systems as quantum simulators for long-range spin-spin interactions, stimulating research is also done in the description of the state of matter for ultracold bosonic dipolar atoms in optical traps. For such frameworks effective bosonic lattice Hamiltonians can be derived [70–75]. These effective Hamiltonians may have ground states which are either symmetric with respect to the symmetries of the Hamiltonian (e. g., uniform Mott insulators [76]) or display diagonal (e. g., density wave insulators [75]) or off-diagonal (e. g., superfluids [76]) long-range order or both (e. g., supersolids [75]). We follow the terminology of diagonal and off-diagonal long-range order introduced by Matsuda and Tsuneto [77]. Especially two-dimensional long-range interacting bosonic systems remain problems which are hard to tackle, as in most cases even the crystalline phases occuring in the so-called atomic limit (see Sec. 3) are unknown.

The main problems in treating long-range interacting models numerically are: On the one hand, the fact that all degrees of freedom that interact, couple to the other degrees of freedom. This makes, for example, the simulation of the classical dynamics of long-range interacting particles especially hard [11]. It impacts as well the simulation of lattice models in the classical and quantum regime and a lot of additional effort is needed in order to make algorithms developed for short-range interacting models also work for long-range interactions [24, 30, 31, 34, 35, 78, 79]. On the other hand, nearly all numerical methods require the definition of a computational unit cell or boundary conditions for simulations on finite systems. For long-range interactions the choice of the right calculation geometry is especially crucial as due to the long-range nature of the pair interactions every site is connected to the boundaries and therefore feels the boundary effects.

In this work we systematically investigate the energy of long-range interacting lattice systems which have an arbitrary diagonal long-range ordered state. We describe a protocol in order to check all possible unit cells up to a given extent in order to systematically identify the energetically most beneficial diagonal order. We introduce resummed couplings in order to better approach the thermodynamic limit on the finite unit cells. For Hamiltonians which are diagonal in the local density basis, this procedure makes it possible to evaluate the true energy of a periodic state in the thermodynamic limit, by analysing a unit cell of the state. The focus of this work lies on this diagonal case. Nonetheless, our protocol provides a solid framework for a mean-field calculation or a strong-coupling expansion [80] where one can systematically incorporate quantum fluctuations.

This paper is structured as follows: We introduce our method in Sec. 2, focusing on the determination of all unit cells in Sec. 2.1, introducing the resummed couplings in Sec. 2.2 and describing a discrete steepest descent minimisation in Sec. 2.3. In Sec. 3 we apply the method

to the atomic limit of the extended Bose-Hubbard model on a triangular lattice which can be used to describe a trapped dipolar gas. We determine phase diagrams at a filling of $f = 1/2$ in Sec. 3.1 and $f = 1$ in Sec. 3.2. In Sec. 4 we apply our approach to investigate the ground state of the antiferromagnetic long-range Ising model on the triangular lattice and establish the plain stripe phase to be the ground state of the system for long-range interactions. In Sec. 5 we investigate the classical limit of the Fendley-Sengupta-Sachdev model [81], which can be used to describe an array of Rydberg atoms. We determine the phase diagram for the kagome lattice in Sec. 5.1, and for the lattice with the sites on the links of a kagome structure in Sec. 5.2. We conclude this paper with a summary and outlook in Sec. 6.

## 2 Systematic Evaluation of Diagonal Orders

We describe an approach to systematically investigate the occuring patterns that bosons form on a lattice in the presence of long-range density-density interactions in the atomic limit. To simplify the illustration and notation we restrict ourselves to two-dimensional systems in the formulation of the idea, but generalisations to other dimensions are straightforward.

The bosonic Hamiltonians of interest are of the form

$$H = -\mu \sum_i n_i + \frac{U}{2} \sum_i n_i(n_i - 1) + \frac{V}{2} \sum_{i \neq j} \frac{1}{|\vec{r}_i - \vec{r}_j|^\alpha} n_i n_j, \tag{1}$$

with the bosonic density operators $n_i$ at lattice site $i$, the strength of the chemical potential $\mu$, the density-density coupling strength $V$, and the onsite repulsion strength $U$. The density-density interaction is between all lattice sites, with the strength decaying algebraically with the distance between the sites $|\vec{r}_i - \vec{r}_j|^{-\alpha}$. Here, $\alpha$ is called the algebraic decay exponent and we assume $\alpha$ to be larger than the dimension of the lattice $d$, which corresponds to so-called weak long-range interactions [5]. We will not consider the regime of strong long-range interactions $\alpha \leq d$, in which the common definitions of internal energy and entropy are not applicable and standard thermodynamics breaks down [1–5]. The limit of nearest-neighbour interactions is recovered for $\alpha = \infty$. Regarding $U = \infty$ is called the limit of hardcore bosons, as site occupancies with more than one boson are energetically forbidden. The Hamiltonian can also be considered in a canonical scheme with a certain fixed overall filling $f$. In that case the chemical potential term is an irrelevant constant.

The overall idea of the proposed approach is to examine the energies of relevant ordering patterns on all distinct unit cells up to a certain extent [82]. The unit cells are generated systematically such that all possible unit cells of ordering patterns up to a certain size are considered. The underlying idea of the method is the fact that we can explicitly evaluate the energy of a periodic ordering pattern on its unit cells using resummed couplings even for long-range interactions. An example for the resumming of couplings is depicted in Fig. 1. An in depth description of the details on how to examine the relevant unit cells and how to evaluate the effective resummed couplings as well as the form of the resulting resummed Hamiltonian is provided in Sec. 2.1 and Sec. 2.2.

For small unit cells all possible occuring states can be evaluated such that the global minimum of the effective Hamiltonian on the unit cell can be found explicitly. For larger translational unit cells we introduce a global optimisation scheme with a local discrete steepest descent optimisation to find the optimal ordering pattern (see Sec. 2.3).

In the end the energies of the ordering pattern on each unit cell with the lowest energy are compared and the overall energetically lowest ordering pattern is considered the ground state of the system in the thermodynamic limit.

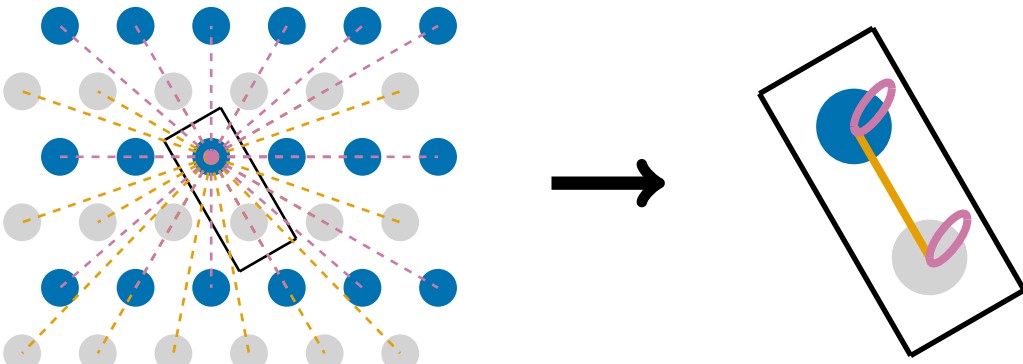

Figure 1: Illustration of the scheme of absorbing interactions into effective resummed interactions. The example presented is a stripe order unit cell on the triangular lattice. The left hand side illustrates the full lattice in the thermodynamic limit with blue circles ● indicating lattice sites with a certain occupation and light gray circles ● indicating lattice sites with another occupation. The right hand side illustrates the unit cell on which the energies of all occuring orders which are contained within that unit cell can be evaluated. The dashed purple lines on the left are absorbed into the resummed single-site density-density interactions on the right hand side while the dashed orange lines are absorbed into the resummed inter-site density-density interaction on the right hand side.

## 2.1 Generation of Unit Cells

We start by presenting an algorithm to determine the distinct unit cells for an arbitrary lattice. At the beginning of our consideration there is an arbitrary lattice with $m$ sites per elementary unit cell

$$\mathcal{L}(\vec{t}_1, \vec{t}_2; \vec{\delta}_1, ..., \vec{\delta}_m) = \{p\vec{t}_1 + q\vec{t}_2 + \vec{\delta}_k | p, q \in \mathbb{Z} \land k \in \{1, ..., m\}\}, \tag{2}$$

with $\vec{t}_1$ and $\vec{t}_2$ being the translational vectors of the elementary unit cell and $\vec{\delta}_k$ the positions of the sites within the elementary unit cell. The translation lattice, that is spanned by the translational vectors, is given by

$$\mathcal{L}(\vec{t}_1, \vec{t}_2; \vec{0}) = \{p\vec{t}_1 + q\vec{t}_2 | p, q \in \mathbb{Z}\}, \tag{3}$$

and the original lattice can be recovered from the lattice spanned by the translational vectors by adding the position vectors of the lattice sites within a unit cell to every element of the translation lattice

$$\mathcal{L}(\vec{t}_1, \vec{t}_2; \vec{\delta}_1, ..., \vec{\delta}_m) = \{\vec{t} + \vec{\delta}_k | \vec{t} \in \mathcal{L}(\vec{t}_1, \vec{t}_2; \vec{0}) \land k \in \{1, ..., m\}\} . \tag{4}$$

Note that $\mathcal{L}(\vec{t}_1, \vec{t}_2, \vec{0})$ forms a $\mathbb{Z}$-module (for a definition see Appendix A) and is isomorphic to the integer lattice $\mathbb{Z}^2$ for the generic elementwise addition and scalar multiplication with whole numbers under the following map

$$\phi : \mathcal{L}(\vec{t}_1, \vec{t}_2; \vec{0}) \rightarrow \mathbb{Z}^2, \qquad p\vec{t}_1 + q\vec{t}_2 \mapsto (p, q), \tag{5}$$

as $\phi$ is a linear and bijective map. To simplify the problem we can therefore consider the integer lattice $\mathbb{Z}^2$. The task is now to identify all translational z-unit cells of the integer lattice, with the translational z-vectors being limited to a given size. We introduce the terminology of translational z-unit cells and translational z-vectors in order to make explicitly clear that we refer to objects on the integer lattice.

| Algorithm | Example |
|---|---|
| Consider an arbitrary two-dimensional lattice with $m$ sites per elementary unit-cell | |
| Regard the translational lattice spanned by the translational vectors $\vec{t}_1$ and $\vec{t}_2$ | |
| Map the translational lattice onto the integer lattice | |
| Generate all relevant pairs of translational z-vectors $(\vec{Z}_1, \vec{Z}_2)$ that form a two dimensional lattice | |
| Find representatives $(\vec{Z}_1, \vec{Z}_2)$ for each distinct lattice and regard the respective unit cell | |
| Introduce back the lattice geometry | |
| Evaluate the resummed couplings for each unit cell | |
| Find the energetically most beneficial order for each unit cell | |
| Compare the best orders of all unit cells to obtain the resulting ordering pattern | |

Figure 2: Schematic overview depiction of the approach to determine the periodic ground state of a generic Hamiltonian of the form given by Eq. (1). The pictures on the right hand side show, as a visualisation, the application of the first five steps to the three site per unit cell Kagome lattice.

To generate the independent pairs of translational z-vectors $(\vec{Z}_1, \vec{Z}_2)$ that fit on the integer lattice up to a given extent and form again a two-dimensional lattice, we propose the following algorithm. Consider a subset $I \subset \mathbb{Z}^2$ of the integer lattice in which all $\vec{Z}_i$ of interest are included. Next, consider the set

$$\tilde{I}^2 := \left\{ (\vec{Z}_1, \vec{Z}_2) \,|\, \vec{Z}_1, \vec{Z}_2 \in I \wedge \neg(\exists r, s \in \mathbb{Z}\setminus\{0\} \ni r \cdot \vec{Z}_1 + s \cdot \vec{Z}_2 = \vec{0}) \right\}, \tag{6}$$

which contains all pairs $(\vec{Z}_1, \vec{Z}_2) \in I$ that are a basis of a two-dimensional lattice. However, many of them are redundant as they form the same lattice. We consider two pairs $(\vec{Z}_1, \vec{Z}_2)$ and $(\vec{Z}_3, \vec{Z}_4)$ to be equivalent if they obey the following four conditions

$$\exists r, s \in \mathbb{Z} \ni r \cdot \vec{Z}_1 + s \cdot \vec{Z}_2 = \vec{Z}_3, \tag{7}$$

$$\exists t, u \in \mathbb{Z} \ni t \cdot \vec{Z}_1 + u \cdot \vec{Z}_2 = \vec{Z}_4, \tag{8}$$

$$\exists v, w \in \mathbb{Z} \ni v \cdot \vec{Z}_3 + w \cdot \vec{Z}_4 = \vec{Z}_1, \tag{9}$$

$$\exists x, y \in \mathbb{Z} \ni x \cdot \vec{Z}_3 + y \cdot \vec{Z}_4 = \vec{Z}_2. \tag{10}$$

We can now partition the elements in $\tilde{I}^2$ into equivalence classes and choose one representative for each equivalence class. The set of representatives for each equivalence class provides the desired set of independent pairs of translational z-vectors. A more mathematical formulation of determining the independent pairs of translational z-vectors would be, to determine all submodules of $\mathbb{Z}^2$ with a rank of two which have a basis with basis vectors being elements of $I$. A basis for each of these submodules would be the independent pairs of translational z-vectors in question (see Appendix A).

The next step is to determine the translational z-unit cells of the integer lattice which are translated by the translational z-vectors. Regarding a pair $(\vec{Z}_1, \vec{Z}_2)$ that forms a two-dimensional lattice, we define an equivalence relation for two lattice points $\vec{z}_a, \vec{z}_b \in \mathbb{Z}^2$ of the integer lattice as

$$\vec{z}_a \sim_{(\vec{Z}_1, \vec{Z}_2)} \vec{z}_b := \exists r, s \in \mathbb{Z} \ni \vec{z}_a + r\vec{Z}_1 + s\vec{Z}_2 = \vec{z}_b. \tag{11}$$

Taking the set of representatives $\{\vec{z}_1, ..., \vec{z}_n\}$ and requiring the representatives to be neighbours with at least one other representative provides a set of points which forms a unit cell in the common notion. Having the translational z-unit cells with the translational z-vectors one can invert the isomorphism from the integer to the real translational lattice

$$\phi^{-1}(\vec{Z}_1) = \phi^{-1}((Z_{1,1}, Z_{1,2})) = Z_{1,1}\vec{t}_1 + Z_{1,2}\vec{t}_2 := \vec{T}_1, \tag{12}$$

$$\phi^{-1}(\vec{Z}_2) = \phi^{-1}((Z_{2,1}, Z_{2,2})) = Z_{2,1}\vec{t}_1 + Z_{2,2}\vec{t}_2 := \vec{T}_2, \tag{13}$$

$$\phi^{-1}(\vec{z}_i) = \phi^{-1}((z_{i,1}, z_{i,2})) = z_{i,1}\vec{t}_1 + z_{i,2}\vec{t}_2 := \vec{\Delta}_i. \tag{14}$$

Note that $(\vec{T}_1, \vec{T}_2, \{\vec{\Delta}_1, ..., \vec{\Delta}_n\})$ forms a unit cell on the translational lattice with translational vectors $\vec{T}_1$ and $\vec{T}_2$ with $n$ elementary unit cells of the original lattice at positions $\vec{\Delta}_i$. To obtain the unit cell for the original lattice the last step is to insert the positions within the elementary unit cell

$$\left(\vec{T}_1, \vec{T}_2, \{\vec{\Delta}_1, ..., \vec{\Delta}_n\}\right) \longrightarrow \left(\vec{T}_1, \vec{T}_2, \{\vec{\Delta}_l + \vec{\delta}_k | l \in \{1, ..., n\}, k \in \{1, ..., m\}\}\right). \tag{15}$$

An example of the procedure so far is schematically presented on the right hand side of Fig. 2 for the Kagome lattice.

Note, that we chose the detour over the integer lattice for several reasons. First, it is computationally much easier to deal solely with integers and omit the geometry of the translational lattice as no floating point number operations have to be performed. Second, it is the most general way in the sense that one can do the determination of translational z-vectors and translational z-unit cells once and then construct any desired lattice by reintroducing the lattice geometry.

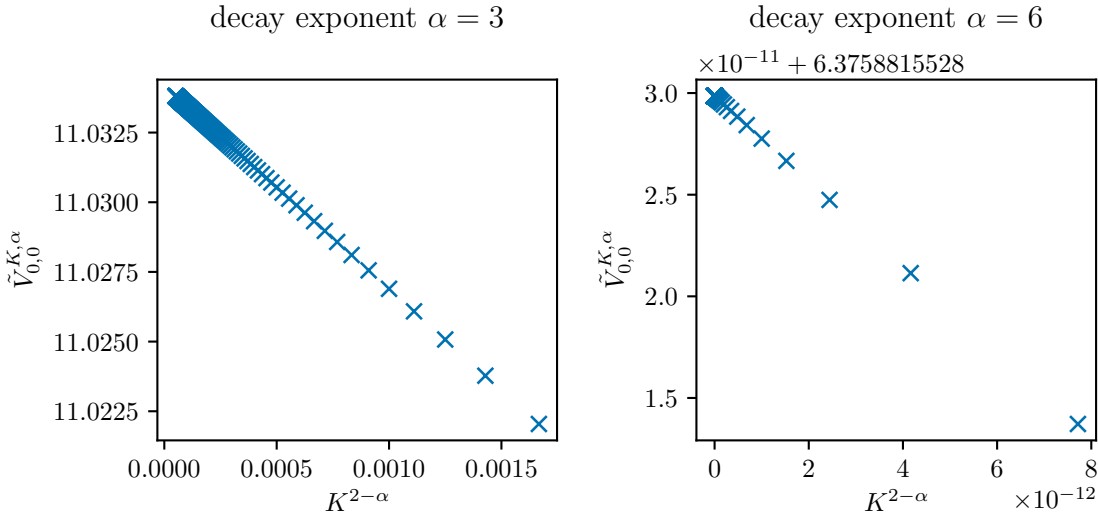

Figure 3: Demonstration of the extrapolation of the resummed couplings assuming the $K^{2-\alpha}$ dependence of the resummed coupling on the cutoff. The plot on the left-hand side demonstrates the extrapolation for $\alpha = 3$ and the plot on the right hand side for $\alpha = 6$. The presented coupling is the self-interaction $\tilde{V}_{0,0}^{K,\alpha}$ for the single site unit cell of the triangular lattice.

## 2.2 Introduction of Resummed Couplings

At this stage we have created the relevant unit cells with points $\vec{r}_1, ..., \vec{r}_p$ of the original lattice and the translational vectors of the unit cell $\vec{T}_1, \vec{T}_2$. From this information resummed couplings treating the long-range interactions can be calculated.

We define the resummed coupling

$$\tilde{V}_{i,j}^{K,\alpha} := V \sum_{k=-K}^{K} \sum_{l=-K}^{K} \frac{1}{|\vec{r}_i - \vec{r}_j - k\vec{T}_1 - l\vec{T}_2|^\alpha} \tag{16}$$

between two sites at positions $\vec{r}_i$ and $\vec{r}_j$ of the unit cell, with translational vectors $\vec{T}_1$ and $\vec{T}_2$, with the decay exponent $\alpha$, and a cutoff for the resummation $K$. For two-dimensional lattice problems where one cannot perform the resummation analytically, it is necessary to introduce the cutoff chosen appropriately such that the series is converged [32]. Note, it is crucial to avoid errors due to floating point precision during the summation. To further improve the convergence of the resummation one could estimate the error on the infinite resummation by integrating the rest of the series. As we are considering a two-dimensional problem we expect a scaling of the errors to the infinite sum with $K^{2-\alpha}$ in a crude approximation.[4] A similar argument has been made for one-dimensional systems [28] and for the two-dimensional square lattice [83]. Knowing the scaling of the errors $\tilde{\epsilon}_{i,j}^{K,\alpha}$

$$\tilde{V}_{i,j}^{\infty,\alpha} \approx \tilde{V}_{i,j}^{K,\alpha} + \tilde{\epsilon}_{i,j}^{K,\alpha}, \tag{17}$$

we can exploit that $\tilde{V}_{i,j}^{K,\alpha}$ is a linear function in $K^{2-\alpha}$ so that the intersection with the $y$-axis represents an estimate for the infinite series [28].

---

[4]Integrating $|x^2 + y^2|^{-\alpha/2}$ starting with a square, a circle or an ellipsis going to infinity all lead to the result that the errors to the resummation scale with $K^{2-\alpha}$.

Table 1: Operations proposed for the discrete steepest descent algorithm.

| Name | Explanation | Framework |
|---|---|---|
| move-$ij$ | moves particle from site $i$ to site $j$ | canonical and grandcanonical |
| insert-$i$ | inserts a particle at site $i$ | only grandcanonical |
| remove-$i$ | removes a particle at site $i$ | only grandcanonical |

Using the resummed and extrapolated couplings $\tilde{V}_{i,j}^{\infty,\alpha}$ we can now reformulate the Hamiltonian in Eq. (1) on the unit cell

$$\tilde{H} = -\mu \sum_i n_i + \frac{1}{2}\sum_{i \neq j}\tilde{V}_{i,j}^{\infty,\alpha}n_i n_j + \frac{1}{2}\sum_i \tilde{V}_{i,i}^{\infty,\alpha}n_i n_i + \frac{U}{2}\sum_i n_i(n_i - 1), \qquad (18)$$

with the sums running over the finite unit cell. Evaluating the energy of an order on a unit cell and dividing it by the number of sites of the unit cell gives the energy per site $\epsilon_0$ of the order in the thermodynamic limit.

Note, the above procedure to determine the resummed couplings $\tilde{V}_{i,j}^{\infty,\alpha}$ is just a proposition used for the results within this paper. This method definitely works for the considered decay exponents $\alpha = 3, 6, 10$ as the interaction decays quickly enough. From our experience regarding $\alpha \lessgtr 2.5$ the direct summation method we propose becomes increasingly difficult to handle. In that case, a more sophisticated resummation scheme will become necessary to treat the tail of the interaction properly.

All the unit cells with the respective resummed couplings considered in this work are provided in Ref. [84].

## 2.3 Minimisation Scheme

Given a unit cell with an effective Hamiltonian of the form in Eq. (18) the question about the energetically most beneficial configuration arises. For small unit cells with only a few lattice sites, it is possible to check the energy of all relevant states systematically by explicit evaluation. This is especially feasible in a canonical scheme with a fixed filling, as the additional constraint can be used for the generation of the states. With increasing unit cell size and increasing occupations it is no longer possible to check the energy of all states. To find an estimate for the energetically most beneficial configuration, we suggest a global minimisation scheme inspired by the Basin-Hopping algorithm [85] with a local optimisation using a discrete form of the steepest descent rule.

The local optimisation is performed as follows. Starting with a state $s_0$ the subsequent routine is repeated:

The move-$ij$ operation is proposed for all pairs of sites $i$ and $j$ and the insert-$i$/remove-$i$ operations are proposed for all sites $i$. Note that in a canonical scheme with fixed filling only move-$ij$ operations are proposed. The energy differences $\Delta E = E_{\text{after}} - E_{\text{before}}$ are evaluated for the operations listed in Tab. 1. If there exist proposals for operations which have $\Delta E < 0$ the one with the largest amplitude of $\Delta E$ is selected and the evaluation of the propositions starts again. If there is no proposal for operations which have $\Delta E < 0$ a local minimum is reached and the optimisation terminates.

After finding a local minimum the starting state of the local optimisation is randomly scrambled using random move-$ij$ operations and the local optimisation is performed for the resulting state. The energy and state of the lowest encountered local minimum is kept track of. After a seemingly sufficient number of local optimisations the global optimisation is terminated and the best local minimum is then considered the global minimum of the optimisation.

We repeat the global optimisation routine until a suggestion for an energy for a global minimum is reached $n = 10$ times without finding a lower global minimum in the meantime. This increases the reliability of the result coming from the global optimisation routine. Of course, this algorithm does not guarantee the finding of the true global minimum, but for practical purposes it suffices. We would like to emphasise that the discussed optimisation scheme is only a suggestion which is used within this work. It is beyond the scope of this work to access different discrete optimisation schemes and develop fast reliable discrete optimisation routines.

The procedure above is employed on every unit cell considered for the same parameters of the original Hamiltonian and then the energetically most beneficial order between the considered unit cells is selected.

With this algorithm we can determine the occuring order and rule out any competing order which would fit onto all the other considered unit cells but does potentially not fit onto the unit cell of the energetically most beneficial order.

## 3 Atomic Limit of a Dipolar Gas Trapped in an Optical Lattice

The Hamiltonian (1) is occurring as the atomic limit of the effective description of an ultracold bosonic dipolar gas trapped in an optical lattice [74, 86]. In the atomic limit all fluctuations between the sites are reduced to a negligible amount and the problem becomes diagonal in the basis of local occupations in real space. A suppression of all hopping terms can be achieved with a very deep optical lattice which would be analogous to increasing the distance between lattice sites to reduce the overlap of the local wave functions, therefore the notion atomic limit. We consider a bosonic dipolar gas with fixed filling $f$ which modifies the generic Hamiltonian in Eq. (1) to

$$H = \frac{V}{2} \sum_{i \neq j} \frac{1}{|\vec{r}_i - \vec{r}_j|^3} n_i n_j + \frac{U}{2} \sum_i n_i(n_i - 1). \tag{19}$$

We considered all unit cells for the orders which have translational vectors out of the set $\mathcal{B}_m$ with $m = 6$ which is defined as

$$\mathcal{B}_m := \{(i, j) \,|\, i \in \{-m, m\}, \; j \in \{\max(-m - i, -m), ..., \min(m - i, m)\}\}. \tag{20}$$

The considered unit cells and resummed couplings are provided in Ref. [84]. We present results for a filling $f = 1/2$ in Sec. 3.1 and $f = 1$ in Sec. 3.2.

### 3.1 Fixed Filling $f = 1/2$

For the filling $f = 1/2$ only unit cells with an even number of sites are considered for the search of ordering patterns, as there cannot be a periodic order with a unit cell that contains an odd number of sites for a filling $f = 1/2$. For small $V/U$ we find plain stripes with sites occupied by a single boson (see Fig. 4). In this regime the energy penalty for stacking bosons is dominant, therefore it is reasonable that no sites are occupied with more than one boson. We point out, that a plain stripe pattern is also the energetically most beneficial ordering of hardcore bosons on the triangular lattice at half filling with long-range interactions which we can confirm rigorously with our approach, as we intrinsically check the unit cells of all other candidates (see Sec. 4).

With increasing $V/U$ the energy penalty for doubly occupied sites deminishes in comparison to the repulsion of bosons. Due to the particular pattern of the plain stripes it becomes energetically beneficial at a certain point to introduce defect lines perpendicular to the plain



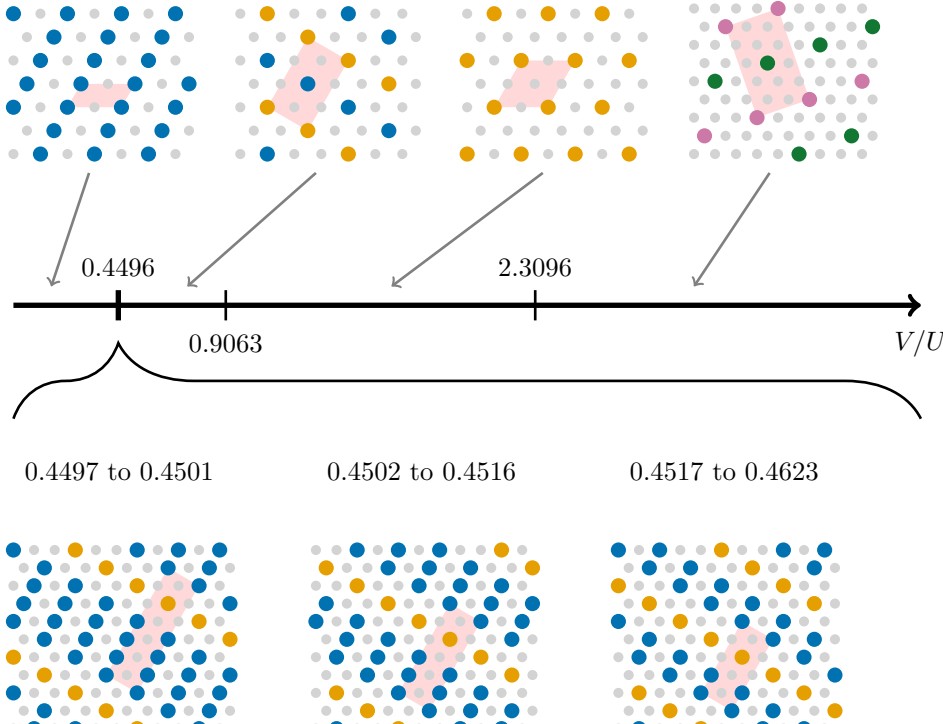

Figure 4: Phase diagram of Hamiltonian (19) at a filling $f = 1/2$ evaluated using the method described in Sec. 2. In the sketches of the phases light gray circles ⚬ indicate empty sites, blue circles 🔵 indicate sites occupied by one boson, orange circles 🟠 indicate sites occupied by two bosons, dark green circles 🟢 indicate sites occupied by three bosons and violet circles 🟣 indicate sites occupied by four bosons. The red-shaded regions indicate unit cells of the respective orders.

stripes with sites occupied by two bosons. A schematic illustration of the creation of defect lines is presented in Fig. 5. Depending on the precise value of $V/U$ those defect lines have a certain distance $d_s \in \mathbb{N}$ between each other and we expect an infinite cascade starting from large $d_s$ towards small $d_s$ for increasing $V/U$ (see Fig. 4). The extent of the orderings with defect lines with a certain $d_s$ in the phase diagram is increasing with $V/U$. The transitions within the staircase are first-order transitions. Note in Fig. 4 only defect lines up to $d_s = 4$ are depicted. We expect every $d_s > 4$ to be realised, but the $d_s = 4$ is the largest defect structure hosted on the investigated unit cells. Regarding the $d_s = 1, ..., 4$ phases, we conclude that the remaining $d_s > 4$ defect patterns have a very rapidly diminishing extent in $V/U$ towards the plain stripe phase. From the numerically accessible small $d_s$ we infer heuristically an algebraically decaying extent with a decay exponent $\gamma \approx 5$. This result might change when regarding $d_s$ over several length scales, which is not feasible with our approach.

At the end of this staircase there is a plain stripe phase with defect lines with distance $d_s = 1$ which is better to think of as an emergent supercrystal of bosons with a pattern of single and doubly occupied sites aligning in a plain stripe-like fashion perpendicular to the single boson occupation plain stripes.

With increasing $V/U$ having bosons neighbouring each other becomes energetically disadvantageous, therefore at $V/U = 0.9063$ there is a first-oder transition to a hexagonal crystal of doubly occupied lattice sites with a distance of two lattice spacings between the occupied sites. The minimal unit cell of this supercrystal has four lattice sites and it is possible to define a unit cell with four elements obeying the symmetries of the triangular lattice.

We find empirically the following overall rule for which superstructures occur (see Tab. 2):

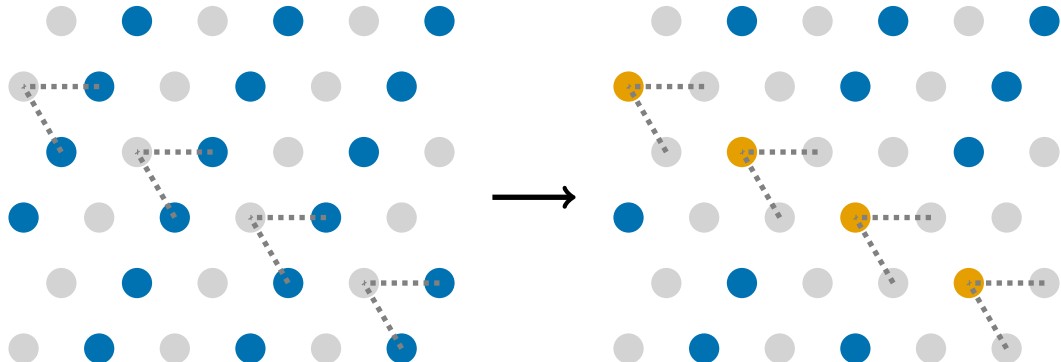

Figure 5: Schematic illustration of the process of creating defect lines with an occupation of two bosons perpendicular to the plain stripe phase. Light gray circles ○ indicate empty sites, blue circles ● indicate sites occupied by one boson and orange circles ● indicate sites occupied by two bosons. The gray dotted lines indicate that a way to understand the occurance of defect lines is by rearranging two bosons from the stripes to a mutual neighbouring site, taking an energetic loss from the onsite repulsion, but benefiting energetically in the density-density interactions from a further spread of the bosons.

Table 2: Phases determined using our method which obey the odd-even rule, including larger $V/U$ than depicted in Fig. 4. The plain stripes, 1,2-Phase, 2-Hexagonal Phase and 3,4-Phase are depicted in Fig. 4. The subsequent phases are generalisations of these patterns to larger occupations with a larger extent.

| $N$ | Phase | Extent $[V/U]$ |
|---|---|---|
| 1 | Plain Stripes | to 0.4496 |
| 3 | 1,2-Phase | 0.4624 to 0.9063 |
| 4 | 2-Hexagonal Phase | 0.9064 to 2.3096 |
| 7 | 3,4-Phase | 2.3097 to 4.0511 |
| 9 | 4,5-Phase | 4.0512 to 5.9540 |
| 12 | 6-Hexagonal Phase | 5.9541 to 8.4147 |
| 13 | 6,7-Phase | 8.4148 to 9.7837 |
| 16 | 8-Hexagonal Phase | 9.7838 to 13.3711 |
| 19 | 9,10-Phase | 13.3712 to ... |

With increasing density-density repulsion the number of sites of successively larger hexagonal unit cells of the triangular lattice becomes relevant. If the number of sites $N$ of the next hexagonal unit cell is odd one expects the next crystalline phase to have two different occupations which are $n_1 = \lfloor N/2 \rfloor$ and $n_2 = \lfloor N/2 \rfloor + 1$. We name those phases as $n_1, n_2$-Phases, e.g., the 1,2-Phase or the 3,4-Phase in Fig. 4. If the number of sites $N$ of the next hexagonal unit cell is even one expects the next crystalline phase to have this unit cell and to have only one site of the hexagonal unit cell occupied with $N/2$ bosons.

We investigated this rule up to $N = 19$ and present our results in Tab. 2. We see that, besides the exception that the defect line staircase pose, the rule is fulfilled. This underlines the special standing of the defect line phases and it is up to future research to assess the behaviour of the defect region when hopping terms and correlated hopping terms are included to the Hamiltonian.

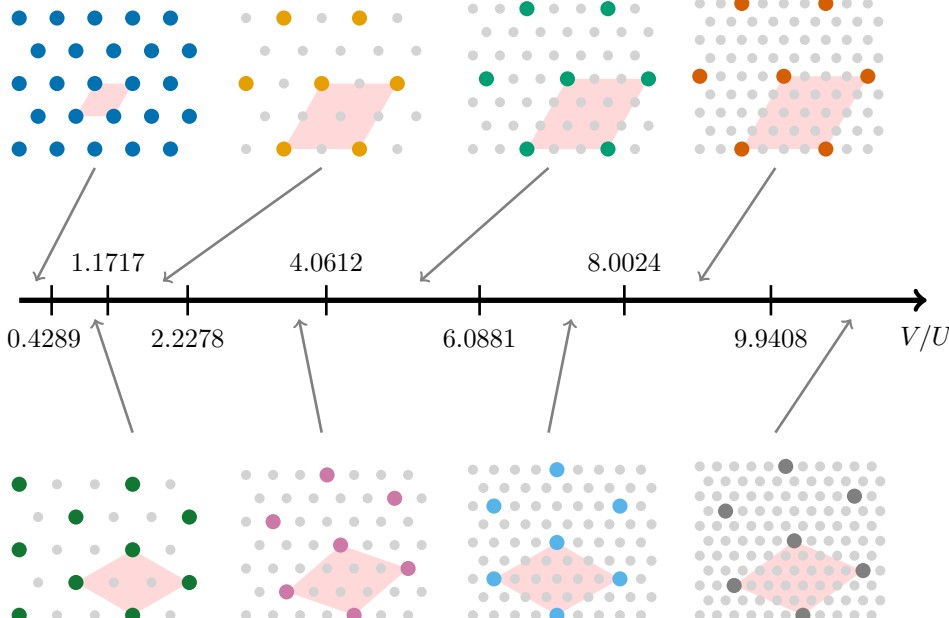

Figure 6: Phase diagram of the Hamiltonian in Eq. (19) with a filling of $f = 1$ evaluated using the method described in Sec. 2. In the sketches of the phases light gray circles ● indicate empty sites, blue circles ● indicate sites occupied by one boson, dark green circles ● indicate sites accupied by three bosons, orange circles ● indicate sites occupied by four bosons, purple circles ● indicate sites occupied by seven bosons, turquoise circles ● indicate sites occupied by nine bosons, sky blue circles ● indicate sites occupied by twelve bosons, red circles ● indicate sites occupied by 13 bosons and gray circles ● indicate sites occupied by 16 bosons. The colour code is the same as in Fig. 7. The red-shaded regions indicate unit cells of the respective orders.

## 3.2 Fixed Filling $f = 1$

For a filling of $f = 1$ unit cells with an even and an odd number of sites are considered for the search of ordering patterns.

A phase diagram is depicted in Fig. 6. For large onsite repulsions a uniform filling of one boson per site is observed. With increasing density-density repulsion larger and larger hexagonal superstructures are observed. Those superstructures have only one single occupation number for occupied sites.

It is an interesting observation that the unit cells of the phases occuring for increasing $V/U$ are precisely the ones of the hexagonal unit cells that cover the whole triangular lattice. We checked this for all realisable unit cells with translational vectors from $\mathcal{B}_{m=6}$ and it indeed holds. Therefore we can predict the sequence of orderings that occur, as we can calculate the hexagonal unit cells and determine the sizes of the unit cells, this is giving us the occupation of the site in the middle of the unit cell and the translational vectors. It is also easily possible to evaluate the energy of such a phase as a function of $V$ and $U$ by using the self interaction $\tilde{V}_{0,0}^{\infty,\alpha}$ for the respective hexagonal cell, which allows to reproduce the phase diagram in Fig. 6. For an order with a distance $d$ between the occupied sites, we obtain the following energy function

$$\epsilon(d, V, U) = \frac{V}{2} \tilde{V}_{0,0}^{\infty,\alpha}(d) \, n(d) + \frac{U}{2} \, (n(d) - 1) \,, \qquad \text{with} \qquad n(d) = d^2 \,. \qquad (21)$$

We depict the evaluated energies in Fig. 7. We confirm that the transitions between the hexagonal orders are simple crossings between the energies of the states which are linear functions

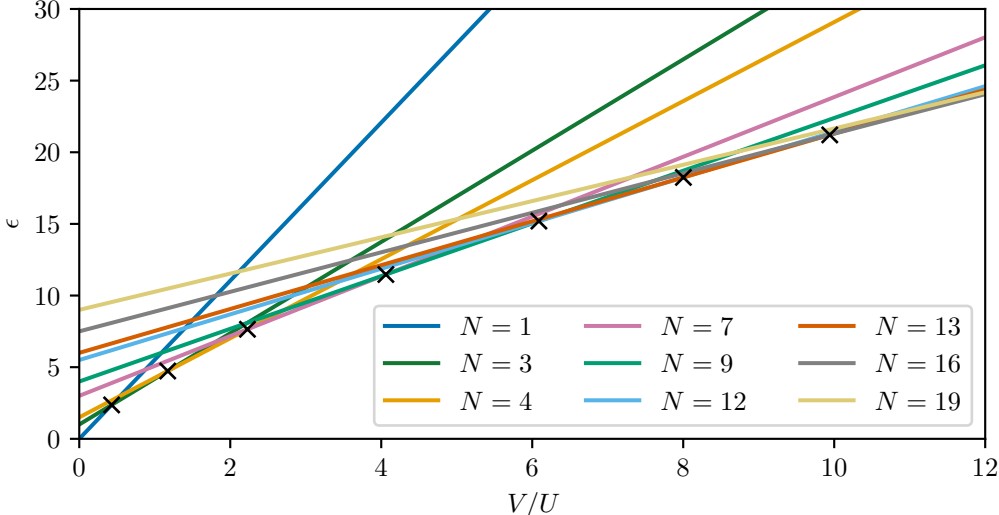

Figure 7: Energies per site of the hexagonal superstructures from 1 to 19 sites per unit cell. The black crosses mark the points where the energies of two superstructures intersect. The colour code is the same as in Fig. 6.

in $V/U$. The observation formalised in Eq. (21) is a great starting point for a simple back of the envelope calculation to roughly determine potentially occuring patterns without the need of the full resummed coupling. One can expand the resummed coupling into its leading contributions

$$\tilde{V}_{0,0}^{\infty,\alpha}(d) = 6\frac{1}{d^\alpha} + 6\frac{1}{(\sqrt{3}d)^\alpha} + 6\frac{1}{(2d)^\alpha} + \dots, \tag{22}$$

and use this appoximation for an estimation of the occuring phase. This approximation is more reasonable for larger $\alpha$ values while for $\alpha = 3$ the full resummed interaction is necessary for a precise determination of the phase boundaries.

It is remarkable that for $f = 1$ as well as for $f = 1/2$ the hexagonal unit cells play a defining role for the occuring phases. When comparing the phase boundaries in Tab. 2 at $f = 1/2$ with the phase boundaries in Fig. 6 at $f = 1$ a qualitative agreement in the transition points between orders associated with the same hexagonal cells is observed. As an example we consider two transitions between a phase associated with the hexagon of size nine and the hexagon of size twelve. The phase boundary between the 4,5-Phase and the 6-Hexagonal Phase is at $V/U = 5.9541$ for $f = 1/2$, while the phase boundary between the phase where a site is occupied by nine bosons and the phase where a site is occupied by twelve bosons is at $V/U = 6.0881$ for $f = 1$. An explanation for this similarity can be obtained using an approximation to determine the energy of the phases at $f = 1/2$. One observes that in the cases of phases with two different occupations both are roughly the same. Therefore instead of distinguishing the different occupations one could apply Eq. (21) with $n(d) = d^2/2$ to account for the half filling. This results in an overall factor of one over two in the energy expression in Eq. (21) which does not change the intersections between the energies.

# 4 Antiferromagnetic Long-Range Ising Model on the Triangular Lattice

As a second application of the approach discussed in Sec. 2 we apply the procedure to the antiferromagnetic long-range Ising model (afLRIM) on the triangular lattice to settle which ordering pattern occurs for $\alpha < \infty$ [26,30,35]. The Hamiltonian of the afLRIM in a transverse field is given by

$$H = \frac{J}{2} \sum_{i \neq j} \frac{1}{|\vec{r}_i - \vec{r}_j|^\alpha} \sigma_i^z \sigma_j^z + h \sum_i \sigma_i^x, \tag{23}$$

with $J > 0$ being the strength of the antiferromagnetic Ising coupling, $\alpha$ the decay exponent of the Ising interaction, and $h$ the transverse-field strength [26, 30, 35]. In the limit $\alpha = \infty$ and $h = 0$ the afLRIM is known to have an extensive ground-state degeneracy which breaks down to a $\sqrt{3} \times \sqrt{3}$ clock order in an order-by-disorder scenario for $h > 0$ [87,88]. The clock order breaks down by a 3DXY quantum phase transition to a trivial paramagnetic high-field phase at $h_c/J = 1.65 \pm 0.05$ [89, 90]. Following the considerations of Humeniuk [91] and Fey et al. [30] for the plain two-dimensional triangular lattice and Saadatmand et al. [35] and Koziol et al. [26] for triangular lattice cylinders this clock-ordered phase remains stable in a finite range of $\alpha < \infty$ for $h > 0$.

Coming from the low-field limit of the afLRIM on triangular lattice cylinders Koziol et al. [26] find very strong indications, that the ground state for small transverse fields is no longer the clock ordered state, but the ground state is adiabatically connected to the ground state of the afLRIM which is found to be a two-fold degenerate plain stripe phase on the cylinders. These stripes are stable under fluctuations induced by the transverse field. Therefore, coming form the limit of small transverse fields on the triangular lattice cylinders, there is first a plain-stripe ordered phase with a first-order phase transition to a clock ordered phase followed by a transition to the trivial high-field phase for $\alpha < \infty$ [26]. Already on the cylinders the choice of the computational unit cell influences the obtained orders, as Saadatmand et al. [35] find a zig-zag striped phase order at low fields while a later investigation by Koziol et al. [26] determines the stripe order to be of a different kind.

Going towards cylinders with a sucessively larger diameter it has been made plausible that the ground state of the afLRIM on the full triangular lattice should be a six-fold degenerate plain stripe ordered phase [26,92,93], but a rigorous numerical proof for this claim is pending.

Using the protocol discussed in this work, we will now treat this problem. The first step is to cast the afLRIM into a hardcore bosonic language using the Matsubara-Matsuda transformation [36] setting $\sigma_i^z = 2n_i - 1$

$$H = \frac{J}{2} \sum_{i \neq j} \frac{1}{|\vec{r}_i - \vec{r}_j|^\alpha} \sigma_i^z \sigma_j^z \tag{24}$$

$$= \frac{J}{2} \sum_{i \neq j} \frac{1}{|\vec{r}_i - \vec{r}_j|^\alpha} (2n_i - 1)(2n_j - 1) \tag{25}$$

$$= \underbrace{2J}_{\frac{\bar{V}}{2}} \sum_{i \neq j} \frac{1}{|\vec{r}_i - \vec{r}_j|^\alpha} n_i n_j - \underbrace{2J}_{\frac{\bar{V}}{2}} \underbrace{\left[ \sum_{j \neq 0} \frac{1}{|\vec{r}_j|^\alpha} \right]}_{\bar{\mu}^\alpha} \sum_i n_i + C \tag{26}$$

$$= \frac{\bar{V}}{2} \sum_{i \neq j} \frac{1}{|\vec{r}_i - \vec{r}_j|^\alpha} n_i n_j - \bar{V} \frac{\bar{\mu}^\alpha}{2} \sum_i n_i + C. \tag{27}$$

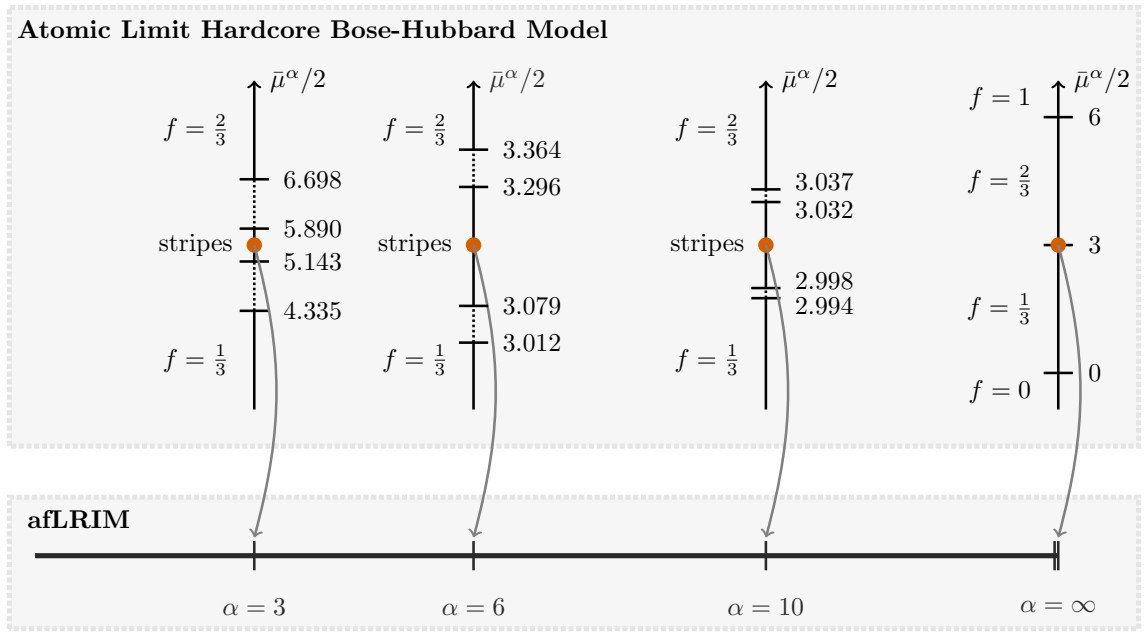

Figure 8: Phase diagram of the afLRIM on the triangular lattice. The ground state of the afLRIM is a sixfold degenerate plain stripe phase for all decay expoents $\alpha < \infty$. The upper part of the figure shows the phase diagram of the atomic limit of the extended hardcore Bose-Hubbard model (see Eq. (27)). In the upper part of the figure $f = 1$ ($f = 0$) indicates the fully filled (empty) solid phase, while $f = 2/3$ and $f = 1/3$ indicate the diagonal orders with the respective filling discussed in references [94, 95]. The expression „stripes" indicates a region in the phase diagram where plain stripes are realised. Dashed lines indicate a cascade of different diagonal orderes which arise due to the long-range interactions but which are not relevant for the understanding of the physics of the afLRIM. The red cricles indicate the parameter values of $\bar{\mu}^{\alpha}$ onto which the afLRIM is mapped (see Tab. 3).

Table 3: Listing of the $\bar{\mu}^{\alpha}$ values the afLRIM is mapped onto for the $\alpha$ values of interest.

| $\alpha$ | $\infty$ | 10 | 6 | 3 |
|---|---|---|---|---|
| $\bar{\mu}^{\alpha}$ | 6 | 6.03144 | 6.37588 | 11.03418 |

We introduce the repulsion strength $\bar{V}/2 = 2J$ and the decay exponent dependent chemical potential $\bar{\mu}^{\alpha}$. The expression in Eq. (27) is now treatable directly by our approach.

An interesting observation can be made regarding the nearest-neighbour limit. Here, it is known that there are extensively many ground states for the afLRIM [87, 88]. The Hamiltonian in Eq. (27) takes the form

$$H_{\alpha=\infty} = \bar{V} \sum_{\langle i,j \rangle} n_i n_j - 3\bar{V} \sum_i n_i, \qquad (28)$$

with a sum over nearest neighbours $\langle i, j \rangle$ and $\bar{\mu}^{\alpha=\infty} = 6$. It is known that the Hamiltonian in Eq. (28) describes a system at a first-order phase transition between a solid phase with fractional filling $f = 1/3$ and a solid phase with fractional filling $f = 2/3$ [94, 95].

In the phase diagram presented in Fig. 8 we demonstrate the behaviour of the afLRIM for $\alpha < \infty$. We investigated the phase diagrams of the Hamiltonian (27) in the vicinity of the $\bar{\mu}^{\alpha}$ values the afLRIM is mapped onto. A listing of the $\bar{\mu}^{\alpha}$ values the afLRIM is mapped onto

for respective $\alpha$ values is provided in Tab. 3. Regarding the phase diagrams of the hardcore bosonic model in Fig. 8 we observe that in the vicinity of the value $\bar{\mu}^{\alpha}$ which the afLRIM is mapped onto, a parameter region in which the stripe ordered phase is stabilised opens up for $\alpha < \infty$. This parameter region is growing in size for smaller $\alpha$ values and is pushed towards higher chemical potentials. We stress that the afLRIM is always mapped to the middle of the plain stripe region for $\alpha < \infty$.

To conclude this section we summarise the observed scenario regarding the ground-state phase diagram of the afLRIM. Mapping the spins onto hardcore bosons we see that for $\alpha = \infty$ the spin model is mapped onto a point of a first order phase transition in the hardcore bosonic Hamiltonian. For $\alpha < \infty$ a stripe phase is stabilised in the hardcore bosonic Hamiltonian and the respective spin models are mapped onto parameters of the hardcore bosonic Hamiltonian that lie within this stripe ordered phase.

As a side note, we remark that the findings for the extended Bose-Hubbard model without hopping at $\alpha = 6$ in Fig. 8 can be compared to experimental data from Rydberg-atom analog quantum simulations and matrix-product state calculations [96, 97]. Regarding the phase diagram presented in Ref. [97] we see a good agreement for the extent of the $f = 1/3$ and $f = 2/3$ phase and the region in between the two phases. We also compared the low density patterns for small $\bar{\mu}^{\alpha}$ with patterns optained in Ref. [59] for a similar regime and find the same prominent density wave orders.

# 5 Classical Limit of Rydberg Atom Arrangements

Recently quantum simulators based on Rydberg atoms were used to investigate $\mathbb{Z}_2$ quantum spin liquid states on the Kagome lattice with sites on the links [65]. Theoretical propositions for states with a topological order were discussed for the site [98] and the link [65, 99] Kagome lattice. The underlying Hamitlonian used to describe the laser-driven Rydberg atom arrays is the so-called Fendley-Sengupta-Sachdev model [63, 65, 81, 98, 99]

$$H = \frac{\Omega}{2}\sum_i (b_i + b_i^{\dagger}) - \delta \sum_i n_i + \frac{V}{2}\sum_{i,j}\frac{1}{|\vec{r}_i - \vec{r}_j|^{\alpha}}n_i n_j\,, \tag{29}$$

with hardcore bosonic creation ($b_i^{\dagger}$), annihilation ($b_i$), and particle-number operators ($n_i$), as well as $\Omega > 0$ and $\alpha = 6$ for Rydberg atoms. Similar to the description of the trapped dipolar gas (see Sec. 3) a classical limit can be taken of the Fendley-Sengupta-Sachdev model. Our aim is to investigate the classical limit of the model on the site Kagome and the link Kagome lattice using the approach described in Sec. 2. It is of great relevance to understand and explicitly demonstrate which diagonal ordering patterns are stabilised by the complete long-range interaction, as often only Rydberg-blockade models or truncated long-range interactions are used in the consideration of the Fendley-Sengupta-Sachdev model. On the other hand, the model is often treated using density-matrix renormalisation-group considerations, which are performed on cylinder geometries [98, 99]. Therefore an investigation of the full long-range interaction in the classical limit helps to assess if diagonal orders of the full long-range interaction are missed in these approximations. In Sec. 5.1 we provide a phase diagram for the model on the site Kagome lattice and in Sec. 5.2 we provide a phase diagram for the model on the link Kagome lattice.

## 5.1 Site Kagome Lattice

In this section we investigate the classical limit of the Fendley-Sengupta-Sachdev model Eq. (29) with $\alpha = 6$ on the site Kagome lattice. We define the lattice with three atoms per unit



Figure 9: Phase diagram of the classical limit of the Fendley-Sengupta-Sachdev model on the Kagome lattice evaluated with a grid of 0.01 in $\delta/V$ and illustrations of the occuring orders with a large extent in $\delta/V$. In the illustrations of the orders the light gray circles ● indicate empty sites and the blue circles ● indicate sites occupied by a boson. The red-shaded regions indicate unit cells of the respective orders. Between the shown phases we expect a devil's staircase of patterns realising inbetween densities. In the chosen grid we encounter such orders, which are indicated by the gray arrows.

cell in the fashion of Eq. (2) in the following way

$$\vec{t}_1 = (2,0)^T \,, \tag{30}$$

$$\vec{t}_2 = \left(1,\sqrt{3}\right)^T \,, \tag{31}$$

$$\vec{\delta}_1 = (0,0)^T \,, \tag{32}$$

$$\vec{\delta}_2 = \frac{1}{2}\left(1,\sqrt{3}\right)^T \,, \tag{33}$$

$$\vec{\delta}_3 = \frac{1}{2}\left(-1,\sqrt{3}\right)^T \,, \tag{34}$$

with $\vec{t}_1$ and $\vec{t}_2$ the translation vectors of the elementary unit cell and $\vec{\delta}_i$ the positions of the three sites within the elementary unit cell.

We evaluated a phase diagram in $\delta/V$, which we present in Fig. 9. We investigated all unit cells with translational vectors out of the set $\mathcal{B}_6$ (see Eq. 20) [84]. The results we present in Fig. 9 are on a grid of 0.01 in $\delta/V$. This is sufficient for the most relevant ordering patterns which occur in the system. As the Hamiltonian is investigated in a grand canonical scheme there is always an infinite staircase of larger ordering patterns in between the phases presented in Fig. 9 and one can easily access them by looking between the phases with a finer grid. We chose not to display these staircases between the grid points as the found orders are on sucessively larger unit cells with a sucessively smaller extent, which eventually leads to limitations in the choice of the unit cells and problems with the visualisation of the patterns. For a one dimensional chain the formation of the infinite staircase is discussed in Ref. [100]. Note that the site Kagome lattice has three sites per elementary unit cell, therefore the biggest clusters for the set $\mathcal{B}_6$ investigated here have $36 \cdot 3 = 108$ sites.

We can directly see that for $\delta \leq 0$ the system realises a completely empty phase as there is no energy gain, or even an energy loss, for adding particles to the system and the energy penalty from the repulsion. As demonstrated in Sec. 4 we can map the classical limit of the hardcore bosonic model onto an afLRIM. In comparison to Sec. 4 we get a longitudinal field for chemical potentials $\delta/V \neq \bar{\mu}^\alpha/2$ (see Eq. 26 for a definition of $\bar{\mu}^\alpha$). With a Matsubara-Matsuda transformation defined as $\sigma_i^z = 2n_i - 1$ we see that the empty state corresponds to the state where all spins have eigenvalue minus one and the filled state to the state with eigenvalues plus one. The states of the Ising Hamiltonian in a positive longitudinal field can be associated with the states of the Hamiltonian in a negative field by a local spin-flip transformation. This manifests in the hardcore bosonic model as a particle-hole symmetry around $\delta/V = \bar{\mu}^\alpha/2$ and imples that the filled state is realised for $\delta/V \geq \bar{\mu}^\alpha$. For the site Kagome lattice with $\alpha = 6$ we calculate $\bar{\mu}^\alpha = 4.283795418$.

Note that the long-range interaction with $\alpha = 6$ is quite weak, therefore the phases from the nearest-neighbour limit have large extent in $\delta/V$. In the nearest-neighbour case the $f = 1/3$ pattern is occuring for $\delta/V \in (0,2)$ and the $f = 2/3$ pattern is occuring for $\delta/V \in (2,4)$ [101]. We observe that the phases with $f = 1/3$ and $f = 2/3$ filling are the most extended in our long-range consideration. Other ordering patterns occur only in small regions between these dominant phases.

Coming from $\delta/V = 0$ and increasing the chemical potential the first ordering pattern we encounter in our grid is a $f = 1/9$ plateau. Of course, in between there is a staircase of other plateaus transitioning from the empty to the $f = 1/9$ plateau. The next plateaus with larger extents are the $f = 1/6$ plateau, then the $f = 2/9$ plateau, before ending up in the $f = 1/3$ phase for $\delta/V \geq 0.11$. In between the $f = 1/3$ and $f = 2/3$ phase we also observe new plateaus to stabilise. First, a $f = 4/9$ pattern is found, then a $f = 1/2$ pattern. Within the $f = 1/2$ pattern the value $\delta/V = \bar{\mu}^\alpha/2$ for the particle-hole symmetry is found, therefore the subsequent phases are the ones already mentioned with particles and holes exchanged.

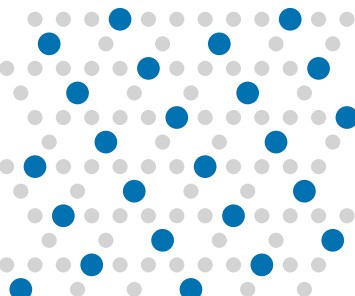 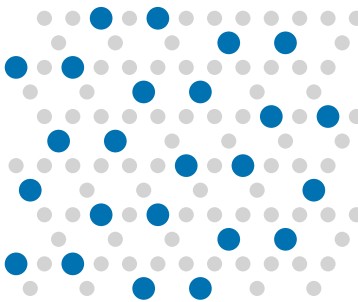

Figure 10: Patterns with a filling $f = 2/9$ on the site Kagome lattice. On the left hand side we depict the pattern that is energetically favoured considering the full long-range interaction. On the right hand side the pattern is depicted which is favoured according to Ref. [98] for truncated long-range interactions after the third-nearest neighbours by an order-by-disorder mechanism for $\Omega > 0$.

In the chosen grid we find multiple of the intermediate patterns. We find a $f = 3/5$ pattern at $\delta/V = 2.20$, a $f = 29/36$ pattern at $\delta/V = 4.22$ and a $f = 11/12$ pattern at $\delta/V = 4.28$. We consider these phases to be less significant as their extent in $\delta/V$ is at least one order of magnitude smaller than the phases depicted in Fig. 9. The fact that we encounter these phases at all is just by the coincidence, that the intermediate densities between the main ordering patterns happen to lie at the points of our grid. For each of these less significant phases we can also find the particle-hole exchanged counterpart by investigating the transitions between the phases with a finer grid. By taking a one order of magnitude smaller grid, we can, for example, find for $\delta/V = 0.003$ and $\delta/V = 0.004$ a $f = 1/12$ phase which is the particle-hole exchanged counterpart of the $f = 11/12$ pattern.

This concludes the explanation of our findings summarised in Fig. 9. We will now use our results to complement and compare to the findings of Samajdar et al. [98] who investigated the full Fendley-Sengupta-Sachdev model with a truncated interaction after the third nearest-neighbours using density-matrix-renormalisation group calculations on finite cylinder geometries. In that work the authors focus on the parameter regime between the $f = 1/6$ and $f = 1/3$ phase. The authors identify in the classical limit three relevant fillings for their consideration $f = 1/6$, $f = 2/9$ and $f = 1/3$. For $f = 1/6$ and $f = 1/3$ the authors identify the same ordering patterns as we find using our approach (see. Fig. 9). As the authors use a trucated interaction, they find a degenerate ground-state space for $f = 2/9$. It is not possible to energetically distinguish between plain strings like in Fig. 9 or strings with kinks (see Fig. 10) having a truncation in the third nearest neighbours. As we take into account the whole long-range interaction using our approach, we see that this degeneracy is broken by the complete long-range interaction and the order depicted in Fig. 9 is prefered. Comparing the phase boundaries of the $f = 1/6$, $f = 2/9$ and $f = 1/3$ orders with Ref. [98] we see a good agreement with very minor deviations due to the full long-range interaction considered in our approach.

Following the arguments in Ref. [98] there is an order-by-disorder scenario when turning on the $\Omega$ term and second-order perturbation theory selects the states with the highest number of kinks out of the degenerate ground-state space with $f = 2/9$ for truncated interactions (see Fig. 10). With their finite cylinder density-matrix renormalisation group approach the authors of Ref. [98] do not find such a state, but strings wrapping around the cylinder circumference. It seems worth to mention, that the occurance of orders wrapping around a cylinder despite an order-by-disorder scenario was also observed for transverse-field Ising models with long-range interactions on the triangular lattice cylinder geometries [26, 35, 92, 93].

We explicitly compared the energy of the state that is favoured for $\Omega > 0$ and the stripe pattern and see that the long-range interaction creates an energy gap between those states. We quantify the energy difference per site between the two states depicted in Fig. 10 to be $\Delta \epsilon^{(0)}(V) = 9.5232 \times 10^{-6} V$. We conjecture a similar scenario as it is expected for the triangular lattice where the full long-range interaction favours precisely the opposite type of states as the order-by-disorder mechanism and for small finite $\Omega$ there is a stable crystaline pattern as depicted in Fig. 9 before entering an order-by-disorder dominated regime at larger $\Omega$ values [26, 35, 92, 93]. Due to the gapped nature of the stripe pattern we find with the full long-range interaction, there has to be a finite region in $\Omega > 0$ where this pattern is stable. The perturbative energy corrections contribute only in even pertubation orders.

## 5.2 Link Kagome Lattice

In this section we investigate the classical limit of the Fendley-Sengupta-Sachdev model Eq. (29) with $\alpha = 6$ on the link Kagome lattice. We define the lattice with six sites per unit cell in the fashion of Eq. (2) in the following way

$$\vec{t}_1 = (0, 4)^T \,, \tag{35}$$

$$\vec{t}_2 = \left(2, 2\sqrt{3}\right)^T \,, \tag{36}$$

$$\vec{\delta}_1 = \frac{1}{2}\left(1, \sqrt{3}\right)^T \,, \tag{37}$$

$$\vec{\delta}_2 = \frac{1}{2}\left(-1, \sqrt{3}\right)^T \,, \tag{38}$$

$$\vec{\delta}_3 = \left(0, \sqrt{3}\right)^T \,, \tag{39}$$

$$\vec{\delta}_4 = \frac{1}{2}\left(-1, -\sqrt{3}\right)^T \,, \tag{40}$$

$$\vec{\delta}_5 = \frac{1}{2}\left(1, -\sqrt{3}\right)^T \,, \tag{41}$$

$$\vec{\delta}_6 = \left(0, -\sqrt{3}\right)^T \,, \tag{42}$$

with $\vec{t}_1$ and $\vec{t}_2$ the translation vectors of the elementary unit cell and $\vec{\delta}_i$ the positions of the six sites within the elementary unit cell.

We present the phase diagram in $\delta/V$ in Fig. 11. We investigate all unit cells with translational vectors out of the set

$$\mathcal{A}_4 = \{(i, j) | i \in \{-4, 4\}, j \in \{-4, 4\}\} \,. \tag{43}$$

The considered unit cells with the resummed couplings are provided in Ref. [84].

By choosing a grid of 0.01 for $\delta/V$ we can determine the phases with larger extent in $\delta/V$ as well as some intermediate phases. As for the site Kagome lattice, in between phases with large extent, there is always an infinite staircase of phases with larger ordering patterns which can be accessed by choosing a finer grid. We do not display patterns where the ordering exceeds the size of the lattice patches chosen for the visualisation.

As already described in Sec. 5.1 the classical limit of the hardcore bosonic model in Eq. (29) can be mapped onto the afLRIM in a longitudinal field. This statement holds independent of the underlying lattice. Here we obtain $\bar{\mu}^{\alpha} = 2.126331592$ for $\alpha = 6$. Note that the phase diagram is symmetric under the exchange of particles and holes around $\bar{\mu}^{\alpha}/2$.

Figure 11: Phase diagram of the classical limit of the Fendley-Sengupta-Sachdev model on the link Kagome lattice evaluated with a grid of 0.01 in $\delta/V$ and illustrations of the most interesting occuring orders. In the illustrations of the orders the light gray circles ○ indicate empty sites and the blue circles ● indicate sites occupied by a boson. The red-shaded regions indicate unit cells of the respective orders. Between the shown phases we expect a devil's staircase of patterns realising inbetween densities. In the chosen grid we encounter such orders, which are indicated by the gray arrows. In the $f = 2/9$ phase, orange circles ● indicate monomer vertices in a dimer-monomer-picture.

For $\delta/V \leq 0$ there are no particles in the lattice as there is no energetic benefit from adding particles and a repulsive interaction. Increasing the chemical potential from $\delta/V = 0$ the first extended phase we encounter is for $f = 2/9$. Then a phase with a filling of $f = 1/4$. This phase is followed by a $f = 1/3$ phase, which is one of the two phases with the largest extent in $\delta/V$, and a $f = 1/2$ phase. Due to the particle-hole symmetry of the classical limit of the Hamiltonian in Eq. (29) around the value $\delta/V = \bar{\mu}^{\alpha}/2$ which lies within the $f = 1/2$ plateau, the following phases are given by the already encountered ones by exchanging particles and holes. The system finally realises a completely filled phase with $f = 1$ for $\delta/V \geq 2.13$ in accordance with the calculated value of $\bar{\mu}^{\alpha}$.

In between these phases with larger extents, we find multiple intermediate phases with smaller extent on the grid we chose. We find a $f = 2/9$ phase for $\delta/V = 0.01$, a $f = 4/9$ phase for $\delta/V = 1.03$, a $f = 16/27$ phase for $\delta/V = 1.10$, a $f = 17/24$ phase for $\delta/V = 2.09$ and a $f = 5/6$ phase for $\delta/V = 2.12$. Looking at the extent of these phases we consider them to be less significant. Using a finer gridding of 0.001 for $\delta/V$ we can find the particle-hole symmetric counterparts of these intermediate phases. For example we find a $f = 5/9$ phase for $\delta/V = 1.095$ to $\delta/V = 1.099$ which is the particle-hole exchanged version of the $f = 4/9$ phase as well as a $f = 11/27$ phase for $\delta/V = 1.026$ which is the particle-hole exchanged version of the $f = 16/27$ phase.

One can switch to a picture where an occupied site on a link of the Kagome lattice corresponds to a dimer connecting the two neighbouring vertices of the lattice. In this picture the region around $f = 1/4$ for $0.02 \leq \delta/V \leq 0.03$ is of particular interest, as the pattern we find for this filling corresponds to a perfect dimer covering where each vertex is touched by exactly one dimer. At fillings smaller than $f = 1/4$ some vertices are not touched by a dimer. Such vertices are referred to as monomers. We show the distribution of monomers exemplarily for the $f = 2/9$ filling pattern in Fig. 11 as orange circles. Using a finer grid with a spacing of 0.001 we further find a $f = 5/18$ phase for $\delta/V = 0.036$.

According to [99] the regime with a dimer-monomer filling with a low monomer density close to $f = 1/4$ is promising to host a quantum spin liquid state for intermediate Rabi frequencies $\Omega$. The authors of Ref. [99] investigate a Rydberg-blockade version of the full model in Eq. (29) where the long-range interaction with $\alpha = 6$ is approximated by forbidding the occupation of the six nearest-neighbours to an already occupied site. Maximal filling in this model corresponds to a perfect dimer covering which highlights the importance of this region in our phase diagram. Note for the Rydberg-blockade model the space of perfect dimer coverings is highly degenerate. We investigate the classical limit of the full model by setting $\Omega = 0$ but include the full long-range interaction. The ordering patterns emerging in this limit pose another starting point from which the system can be understood, especially considering that including longer-range interactions seems to destabilise the quantum spin liquid phase [99].

Following the argumentation of [99, 102–104] for the Rydberg-blockade model, the dimer covering that is chosen out of the degenerate space of dimer coverings by $\Omega > 0$ contains resonating plaquettes which lower the energy in sixth order perturbation theory. Similar to our observations in Sec. 5.1 for the Kagome lattice or in Sec. 4 for the triangular lattice, we see for the dual kagome lattice that the treatment of the entire long-range interaction breaks the degeneracy of dimer coverings. The dimer covering, which we find to have the lowest energy at $\Omega = 0$, is depicted in Fig. 11 and has a dimer alignment without any resonating plaquettes. We quantify the energy difference per site between the state discussed in [99] with 72 sites per unit cell and resonating pinwheel structures and the state we find depicted in Fig. 11 to be $\Delta\epsilon^{(0)}(V) = 4.4283 \times 10^{-5} V$. The relevant interaction that distinguishes the two states energetically is the fifth nearest-neighbour interaction.

Now, we can make similar conjectures as for the order-by-disorder scenario on the site Kagome lattice. As we know that the order we find differs from the pattern that is benefited

the most for $\Omega > 0$ we conjecture a level-crossing transition from the order we find to the valence-bond solid that is discussed in Ref. [99]. We quantify the leading order corrections in $\Omega$ setting $V = 1$ as

$$\Delta\epsilon(\Omega, \delta) = \Delta\epsilon^{(0)} + \Omega^2 \Delta\epsilon^{(2)}(\delta), \qquad (44)$$

with $\Delta\epsilon^{(2)}(\delta = 0.02) = -0.7366516682$ and $\Delta\epsilon^{(2)}(\delta = 0.03) = -2.2644374449$ for the $\delta$ values presented in Fig. 11. We obtain the smallest correction $\Delta\epsilon^{(2)}(\delta = 0.01928) = -0.7313527162$ for $\delta = 0.01928$. By this consideration the stability of the newly obtained state against the pinwheel state can be estimated in leading order to $\sqrt{|\Delta\epsilon^{(0)}/\Delta\epsilon^{(2)}|}$. Using $\Delta\epsilon^{(2)}(\delta = 0.01928)$ we obtain a value of $\Omega = 0.0078$. The results of the second order non-degenerate perturbation theory in $\Omega$ presented can be easily calculated using the resummed couplings used for the evaluation of the $\Omega = 0$ properties. To obtain the fourth order corrections a non-degenerate perturbation theory calculation would have to be performed, but occuring double sums would require additional resummation. In sixth order degenerate perturbation theory is necessary as the plaquettes in the pinwheel states resonate in sixth order and couple the states.

We can also speculate about the observations that long-range interactions seem to obstruct the formation of quantum spin liquid states. We observe that for fillings slightly smaller than $f < 1/4$ in the presence of long-range interactions the monomer sites of the kagome lattice order in a periodic pattern to lower the overall energy (see Fig. 11).

Eitherway, the precise behaviour for $\Omega > 0$ for the whole $\delta/V$ axis and the stability of quantum spin liquids with long-range interactions remain open questions for further research.

# 6 Summary and Outlook

We presented a rigorous approach to investigate diagonal ordering patterns in bosonic lattice models with long-range density-density interactions. The approach is based on a systematical investigation of ordering patterns on all unit cells up to a given size. Here resummed and extrapolated couplings are used to treat the long-range nature of the interaction. In the realm of the work we exclusively focused on diagonal Hamiltonians.

To demonstrate the approach for bosonic systems, we investigated the atomic limit of a Hamiltonian describing an ultracold dipolar atomic gas trapped in a triangular optical lattice for fixed fillings of $f = 1$ and $f = 1/2$ identifying several distinct bosonic superstructures when tuning the onsite repulsion against the density-density interaction strength. For $f = 1$ the system always realises structures with hexagonal unit cells where only a single site is occupied within the unit cell. For $f = 1/2$ we find an overall scheme dependent on the sequence of the sizes of the possible hexagonal unit cells of the triangular lattice to predict occuring crystalline phases. Besides this scheme we find that the plain stripe phase transitions into the 1,2-Phase via a staircase of phases with defect lines. The distance between these defect lines is decreasing towards the 1,2-Phase.

To demonstrate the applicability for spin systems, we investigated the ground-state properties of the antiferromagnetic long-range Ising model. We show that a plain stripe phase is the ground state of the model.

We further present calculations for hardcore bosons with van-der-Waals long-range density-density interactions in a grand canonical scheme on the site and the link Kagome lattice. Recently, Rydberg atom quantum simulators on these lattice geometries were used to simulate quantum spin liquid states [65, 98, 99]. With our approach, we determined the relevant ordering patterns in the limit $\Omega = 0$ of the Fendley-Sengupta-Sachdev model. We find a good agreement with existing considerations with truncated long-range interactions for the site Kagome

lattice [98]. We provide insights on the occuring ordering patterns considering the whole long-range interaction. Similarly, we also discussed findings for the link Kagome lattice with a focus on the dimer covering filling $f = 1/4$.

Now, we would like to draw a path for future applications of the framework: The examples presented in this work focus on solely diagonal problems which correspond to classical optimisations for the energetically best pattern. Nevertheless, the overall framework of this work provides a very easy access to long-range density-density interactions for mean-field calculations. Let us take as an example the hardcore Bose-Hubbard model [82, 94] extended with long-range density-density interactions. The problem can be investigated using mean-field calculations, e.g., the so-called classical approximation [82,94,105]. It is now possible to perform these mean-field calculations for the respective unit cells with resummed density-density interactions. By this, mean-field phase diagrams with non-diagonal terms become computable for the long-range interactions and a whole zoo of phenomena such as supersolids [94,105] or pair superfluids [106] can be investigated in the context of long-range interactions. Further, mean-field studies of soft-core bosons would benefit through the application of our framework [59,66,86]. Related to this, the investigation of metastable states for the full long-range interaction in the atomic limit on the level of mean fields is within reach with our framework [86].

We emphasise that the method introduced here is strictly applicable to weak long-range interactions due to the requirement of an extensive energy in the thermodynamic limit. Nevertheless the extension to strong long-range interactions could be performed by means of Kac scaling. In that case it would be interesting to compare the predictions with the calculations performed for Coulomb interacting systems [107].

# Acknowledgements

JAK thanks Michael Schmiedeberg for fruitful discussions and help with the literature on colloidal systems on a substrate. This work was funded by the Deutsche Forschungsgemeinschaft (DFG, German Research Foundation) - Project-ID 429529648 - TRR 306 QuCoLiMa (Quantum Cooperativity of Light and Matter). KPS acknowledges the support by the Munich Quantum Valley, which is supported by the Bavarian state government with funds from the Hightech Agenda Bayern Plus.

# A    Mathematical Definitions and Theorems

This section contains a more mathematical elaboration on the procedure of determining the unit cells described in Sec. 2.1. We start by creating a framework defining the notion of a $\mathbb{Z}$-module with further important definitions and theorems, specialising the definitions for a general $R$-module to the whole numbers. The defintions presented here are adapted from the textbook „Algebra" by Sergey Lang [108] and the textbook „Algebras, Rings and Modules" [109].

**Definition $\mathbb{Z}$-Module:**   Let $M$ be an Abelian group and there is an operation $\cdot : \mathbb{Z} \times M \longrightarrow M$ such that for all $r, s \in \mathbb{Z}$ and $m, n \in M$ the following four statements hold

$$r \cdot (m + n) = r \cdot m + r \cdot n, \tag{A.1}$$

$$(r + s) \cdot m = r \cdot m + s \cdot m, \tag{A.2}$$

$$(r \cdot s) \cdot m = r \cdot (s \cdot m), \tag{A.3}$$

$$1_{\mathbb{Z}} \cdot m = m, \tag{A.4}$$

then $M$ is called a $\mathbb{Z}$-module and $\cdot$ is called the scalar multiplication.

**Example:**   An example for a $\mathbb{Z}$-module is the $n$-dimensional integer lattice $\mathbb{Z}^n$ with elementwise addition and elementwise scalar multiplication.

**Definition $\mathbb{Z}$-Module Isomorphism:**   Let $M$ and $N$ be $\mathbb{Z}$-modules. A function $f : M \longrightarrow N$ is called a $\mathbb{Z}$-module isomorphism if $f$ is bijective and $\forall m, n \in M$ and $r \in \mathbb{Z}$

$$f(m + n) = f(n) + f(m), \tag{A.5}$$

$$f(r \cdot m) = r \cdot f(m). \tag{A.6}$$

If there exists an $\mathbb{Z}$-module isomorphism between the $\mathbb{Z}$-modules $M$ and $N$, then $M$ and $N$ are called isomorphic.

**Definition $\mathbb{Z}$-Submodule:**   Let $M$ be a $\mathbb{Z}$-module. Let $N$ be a subgroup of the Abelian group $M$. $N$ is called a $\mathbb{Z}$-submodule if for all $n \in N$ and for all $r \in \mathbb{Z}$

$$r \cdot n \in N. \tag{7}$$

**Definition Generator:**   Let $A \subset M$ be a subset of a $\mathbb{Z}$-module. $A$ is called a generator of $M$ if

$$M = \left\{ \sum_{a \in A} z_a \cdot a \, | \, z_a \in \mathbb{Z} \wedge z_a = 0 \text{ for almost all } a \in A \right\}. \tag{8}$$

**Definition Finitely Generated $\mathbb{Z}$-Module:**   A $\mathbb{Z}$-module is called finitely generated if it has a finite generating set.

**Definition Basis:**   A generator $A$ of a $\mathbb{Z}$-module $M$ is called basis of $M$ if the elements of $A$ are linearly independent

$$\sum_{a \in A} z_a \cdot a = 0, \quad \text{with} \quad z_a \in \mathbb{Z} \wedge z_a = 0 \text{ for almost all } a \in A \Rightarrow z_a = 0 \; \forall a \in A. \tag{9}$$

The number of elements in a basis set is called cardinality of a basis.

**Definition Free $\mathbb{Z}$-Module:**   A $\mathbb{Z}$-module is called free if it has a basis.

**Theorem Rank of Free $\mathbb{Z}$-Modules:**   All distinct bases of a free $\mathbb{Z}$-module have the same cardinality. This cardinality is called the rank of a free $\mathbb{Z}$-module.

**Proof:**   See the proof of proposition 1.5.5. in Ref. [109]

**Theorem Isomorphisms to $\mathbb{Z}^n$:** A free finitely generated $\mathbb{Z}$-module with rank $n$ is isomorphic to $\mathbb{Z}^n$.

**Proof:** See the proof of proposition 1.5.3. in Ref. [109]

Now regarding the algorithm in Sec. 2.1. The integer lattice $\mathbb{Z}^2$ is a free $\mathbb{Z}$-module. A basis of $\mathbb{Z}^2$ is $\{(1,0),(0,1)\}$. $\mathbb{Z}^2$ has a rank of two. We want to find all submodules of $\mathbb{Z}^2$ with rank two and a basis with elements of the set $I$ (see Sec. 2.1). These submodules are precisely the lattices which can be formed using points of the original integer lattice and it is secured that these submodules are again two dimensional lattices as these submodules are isomorphic to $\mathbb{Z}^2$. For each of the determined submodules in question a basis is picked and those bases are the distinct translational z-vectors which translate z-unit cells of the integer lattice. This is the mathematical background behind the algorithm and the equivalence relations in Sec. 2.1.

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
