# Peer review of "Systematic Analysis of Crystalline Phases in Bosonic Lattice Models with Algebraically Decaying Density-Density Interactions"

_SciPost Physics, doi:SciPost Phys. 14, 136 (2023)_

## Round 1 · Referee Report · Anonymous (Referee 1) · 2022-12-30

Strengths

1. The authors provide a computational mechanism to identify ground states of quantum systems with long-range potentials in the classical limit where the off-diagonal terms in the computational basis (for example Fock basis in real space) is ignored.

2. The states found using this method could be a good starting point for addressing effect of quantum fluctuations for such systems.

3. The treatment, applied to Rydberg atoms on links of Kagome lattice, finds a somewhat different picture than was found earlier in Ref 97 of the paper. In particular they notice absence of resonatiing plaquettes which may call into question the spin liquid state found earlier. If correct, this is expected to open up further studies in the field.

Weaknesses

1. The authors could have studied effect of the off-diagonal terms on their analysis (perturbatively and numerically) rather than leaving this for a future study. That would have provided a more complete picture.

Report

Overall, the authors provide a computational scheme which may be used to numerically evaluate the ground state configuration of a quantum system with long-range density-density or spin-spin interaction in the classical limit. This method is interesting and may be useful for accurate determination of ground states of such systems. However, they have not estimated the effect of off-diagonal terms ( which are almost always present and important) on their analysis. If this issue can be taken care of (even perturbatively and numerically
for, say, the rydberg atoms on Kagome), the paper shall be much stronger and I would certainly recommend it for publication.

Requested changes

1. It would be nice to have a perturbative and numerical treatment of off-diagonal terms. If this is really impossible, a discussion regarding why that is the case should be included. This is particularly improtant for the Rydberg atoms array on Kagome lattice links.

  • validity: high
  • significance: high
  • originality: high
  • clarity: high
  • formatting: excellent
  • grammar: perfect

Author:  Jan Alexander Koziol  on 2023-03-10  [id 3468]

(in reply to Report 1 on 2022-12-30)

Response: Anonymous Report 1 on 2022-12-30 (Invited)

We thank the referee for examining our work and his suggestions to make the content of our paper stronger.

Below we address the point made by the referee:

1.) We thank the referee for this point. We added a quantitative value for the energy gap between the pinwheel state and the state we find by our consideration in Sec. 5.2. We also added a quantitive value for the leading order corrections in \Omega by which the stability of the newly obtained state against the pinwheel state can be estimated. We added comments on why the evaluation of the fourth and sixth order is impractical or even impossible in the thermodynamic limit. Regarding an unbiased numerical treatment: It seems impossible to us to set up exact diagonalisation calculations due to the large extent of the unit cells of the competing orders. It seems also challenging to perform quantum Monte Carlo simulations. Although a very capable algorithm for the transverse-field long-range Ising model by A. Sandvik (https://doi.org/10.1103/PhysRevE.68.056701) is known, the parameters of interest lie in the regime of small transverse fields (\Omega) in which the algorithm suffers of a severe performance loss due to long auto-correlation times. Also, the presence of the competing antiferromagnetic interactions makes the Monte Carlo sampling less efficient. Lastly, the treatment using tensor network methods of the regime would be a possibility. As we are not very familiar with this type of methods we cannot comment on the possibility, challenges and limitations of applying them.

---

## Round 1 · Referee Report · Anonymous (Referee 2) · 2023-1-31

Strengths

1- This paper provides a systematic algorithm to determine the charge order for lattice classical models with power-law density interactions.
2- Such an analysis provides a good starting point to consider quantum effects (i.e. off-diagonal terms in the Fock basis)

Weaknesses

1- It is limited to weak long-range interactions.
2- There is no study of any quantum case although several such models are mentioned.

Report

I find the paper particularly pedagogical and well written. It is nice to see a systematic method to tackle classical models and find the lowest energy pattern. In fact, it is a bit surprising that such a method has not been proposed before ?
Regarding the motivations, there are several quantum models (and references about them), but in its current form, the paper only consider the classical limit. This would be much more interesting if some precise connections could be made to quantum models.

Requested changes

1- I would suggest to remove lots of quantumness from the abstract and the body of the paper. Indeed, the quantum statistics plays no role in the current study, so that it would apply equivalently to bosons, fermions or classical objects.
2- Is it possible to consider, at least partly, the long-range case as done when using Ewald summation ?
3- I would suggest to remove most references about quantum experiments, quantum phases (superfluids, super solids) which are not relevant to the present study.

  • validity: high
  • significance: high
  • originality: good
  • clarity: top
  • formatting: excellent
  • grammar: excellent

Author:  Jan Alexander Koziol  on 2023-03-10  [id 3467]

(in reply to Report 2 on 2023-01-31)

Response: Anonymous Report 2 on 2023-01-31 (Invited)

We thank the referee for examining our work and the suggestions to improve our paper. We acknowledge the idea of the referee to streamline our paper by removing the discussions about the quantum aspects of the models. Nevertheless, we hope to convince the referee why we would like to keep them as part of our work.

Below we address the points made by the referee:

1.) We thank the referee for this point. We see the argument of the referee that "quantum statistics plays no role in the current study", as the examples chosen consider only the "classical limit" with no offdiagonal terms. But, as we discussed in the description of our method and the outlook of the paper, a straightforward application to mean-field considerations with kinetic terms is possible. We are also convinced that our method will largely benefit the studies of trapped ultracold long-range interacting atomic gases, therefore we would like to keep all the references to the publications in the field. We are convinced of the validity of this proceeding and ensured by the responses of the reports 1 and 3. Additionally we added a perturbative discussion of the leading order offdiagonal effects to Sec 5.2.

2.) We thank the referee for this point. As discussed in our paper, it is relevant for the proposed procedure in the paper that the series leading to the resummed couplings is converging. This makes it, with the chosen definition of long-range interactions, not suitable for strong long-range interactions as in that case the series would not converge. In the case that the series is converging it is, of course, not mandatory to perform the resummation in the same way we do in our approach. In some cases, for example on a one-dimensional chain, there are even closed forms for the resummed couplings which go back to the mathematical discipline of analytic number theory. We choose the described procedure as within the decay exponents of interest alpha=3,6,10 this simple real space summation is quite capable and the errors are controlled. When going to smaller alpha values than three (especially alpha<=2.5), indeed, a more sophisticated resummation scheme will become necessary to treat the tail of the interaction properly. We added a note to the end of Sec. 2.2 elaborating on the limits and the necessity for a potential improvement of our resummation scheme for alpha<=2.5.

3.) We thank the referee for this point. As already discussed in the response to point 1.) raised by the referee, we see a high value of our method to the community of long-range interacting quantum systems. This is also nicely elaborated on by the referee of report 3. We want to stress that the occuring crystalline phases are directly relevant also for the full quantum models as they are gapped phases and, therefore, stable against quantum fluctuations. We added a discussion of the leading perturbative behaviour to Sec. 5.2. In a similar fashion we would argue that occuring supersolid phases often break the translational symmetry in a similar way as occuring solid phases. Therefore, we would see our considerations also relevant for this issue. Concluding our response to this point: We hope to convince the referee that we would like to keep all the references to the publications in the field.

---

## Round 1 · Referee Report · Anonymous (Referee 3) · 2023-2-6

Strengths

1. The work is experimentally relevant, theoretically sound, and the manuscript is also well-written and straightforward to follow.
2. I personally believe that these results, even though classical, would be extremely useful to the ultracold-atoms community because the solid density-wave-ordered states may be regarded as starting points both to benchmark experiments and to incorporate quantum corrections.
3. Moreover, there are often subtle effects from the long-ranged dipolar ($1/r^3$) or van der Waals ($1/r^6$) interactions, which are hard to quantify in simple mean-field calculations, and the careful analysis outlined in this paper bridges this gap.

Weaknesses

Please refer to the attached PDF.

Report

I would recommend publication in SciPost Physics after some revisions (listed in the report).

Requested changes

Please refer to the attached PDF.

Attachment

  • validity: top
  • significance: high
  • originality: good
  • clarity: high
  • formatting: excellent
  • grammar: excellent

Author:  Jan Alexander Koziol  on 2023-03-10  [id 3466]

(in reply to Report 3 on 2023-02-06)

Response: Anonymous Report 3 on 2023-02-06 (Invited)

We thank the referee for thoroughly examining our work and raising several interesting points.

Below we address the points made by the referee:

a) We thank the referee for this point. This is a typographical error which we corrected in Eq. (3), Eq. (4) and Eq.(5).

b) We thank the referee for this point. We added a simple back-of-the envelope calculation to determine the rough phase boundaries for the considered fillings f=1/2 and f=1. Additionally, everybody should be able to easily reproduce the phase boundaries as we will publish our resummed couplings (see point g)). Regarding the extent of the defect line phases dependent on their spread $d_s$, we added to the paper the results of a heuristic analysis regarding the accessible phases using our numerical technique. We see the necessity to have an "effective theory" describing the repulsion of the defect lines, but are currently not able to provide a simple picture.

c) We thank the referee for this point. We added an explanation to the text that a phase with n_1 and n_2 occupations is referred to as n_1,n_2-Phase.

d) We thank the referee for the point and the additional reference. Indeed, we encounter the devil's staircase when studying systems in a grand-canonical scheme. This issue is discussed in Sec. 5 for the site and link Kagome lattice. With our method, we cannot prove rigorously that there exists a devil's staircase, but within the capabilities (limited by the size of the considered unit cells) of our method we encounter intermediate fractions when taking a closer grid between the more prominent fillings. We added the reference to the discussion in Sec. 5.

e) We thank the referee for this point. We adjusted the layout of the figure accordingly.

f) We thank the referee for this point. We added the energy difference between the pinwheel state and the state we find for the full long-range interaction. We also commented on which interaction is the first to distinguish the two states. This is the fifth nearest-neighbour interaction V_5. This is the dominant contribution to the energy gap between the states. Further neighbouring interactions only adjust the energy gap given for the full interaction given in the paper. We also determined the leading order corrections in \Omega to the energy difference between the two states perturbatively.

g) We thank the referee for this point. We agree with the referee that a publication of the code would certainly be a very fruitful step and is our goal for the future. Unfortunately the code is still under development and is not yet ready to publish in its current state. Nevertheless, we will publish the resummed coupling used for our calculations as a data repository on Zenodo. The creation of the resummed couplings is one of the computationally intense part of our numerical evaluation. With this step we hope to encourage people to build on our research and make our results more easily reproducible.

---

## Round 2 · List of Changes

-
At the end of the abstract, a sentence was added to summarise the capabilities of the method.
-
On page 2, we reordered the first paragraph of section 1.
-
On page 2, we rephrased the second paragraph of section 1.
-
On page 2, we enhanced the precision of the statements in the third paragraph by specifically referring to the two-body potential.
-
On page 2, we rephrased the opening sentence of the fourth paragraph of section 1.
-
On page 3, we rephrased the last sentance of the first paragraph of the page.
-
On page 3, we shortened two sentances in of the second paragraph of the page for imporved readability.
-
On page 3, we corrected the statement that ions are trapped in "magneto-optical trapps" at the beginning of the third paragrph of the page.
-
On page 3 and 4, we added a comment on incorporating quantum fluctuations by means of a strong coupling expansion at the end of the second last paragraph of section 1.
-
On page 5, we adjusted the choice of indices in Eqs. (2), (3), (4) and (5) for better consistency.
-
On page 5, we corrected the typographical error in Eqs. (3), (4) and (5) as pointed out by referee 3.
-
On page 7, we adjusted the choice of indices in Eqs. (12), (13) and (15) for better consistency.
-
On page 8, we added two paragraphs at the end of section 2.2. The first one commenting on the limitations of our resummation method, as suggested by referee 2. The second one providing information, that the data used in this work is published in a data repository.
-
On page 10, we added a sentence below Eq. (20) informing that the data used in this work is published in a data repository.
-
On page 11, we added an analysis and discussion of the extends in V/U of the defect line phases at the end of the middle paragraph of the page. This follows the request of referee 3.
-
On page 12 and 13, we added an introduction of the n_1,n_2-Phases in the paragraph splitting between the two sites as requested by referee 3.
-
On page 14 and 15, we added three new paragraphs at the end of section 3.1 including Eqs. (21) and (22) introducing multiple back of the envelope calculations as requested by referee 3.
-
On page 18, we corrected the primitive lattice vector in Eq. (30).
-
On page 19, we changed the order of the graphics in Fig. 10 as requested by referee 3.
-
On page 20, we commented on occuing devil's staicases in the first paragraph of the page. And added the reference for one-dimensional systems suggested by referee 3.
-
On page 21, we added results for the energy gap between the states depicted in Fig. 10 in the last paragraph of section 5.1. as requested by referees 1 and 3. We also added a qualitative discussion of the leading effecty of quantum fluctuations.
-
On page 22, we added a sentence below Eq. (43) informing that the data used in this work is published in a data repository.
-
On page 24, we added results for the energy gap between the states in question in the thrid paragraph of the page as requested by referees 1 and 3.
-
On page 24, we added results of a quantitaive strong-coupling expansion in leading order in a new paragraph aroud Eq. (44) as requested by referees 1 and 3.
-
On page 26, we added an outlook discussing the possibility to apply the proposed method also for strong long-range interactions in the last paragraph of section 6. This follows the comments of referee 2.

---

## Editorial Decision

published